# Elevated Plasma Vitamin B_12_ in Patients with Hepatic Glycogen Storage Diseases

**DOI:** 10.3390/jcm9082326

**Published:** 2020-07-22

**Authors:** Julia Hinkel, Johannes Schmitt, Michael Wurm, Stefanie Rosenbaum-Fabian, Karl Otfried Schwab, Donald W. Jacobsen, Ute Spiekerkoetter, Sergey N. Fedosov, Luciana Hannibal, Sarah C. Grünert

**Affiliations:** 1Department of General Pediatrics, Adolescent Medicine and Neonatology, Faculty of Medicine, Medical Center—University of Freiburg, 79106 Freiburg, Germany; julia.hinkel@uniklinik-freiburg.de (J.H.); johannes.schmitt@uniklinik-freiburg.de (J.S.); stefanie.rosenbaum-fabian@uniklinik-freiburg.de (S.R.-F.); karl.otfried.schwab@uniklinik-freiburg.de (K.O.S.); ute.spiekerkoetter@uniklinik-freiburg.de (U.S.); 2Department of Pediatrics, St. Hedwigs Campus, University Children’s Hospital Regensburg, 93049 Regensburg, Germany; Michael.Wurm@barmherzige-regensburg.de; 3Department of Cardiovascular and Metabolic Sciences, Lerner Research Institute, Cleveland Clinic, Cleveland, OH 44195, USA; jacobsd@ccf.org; 4Department of Molecular Biology and Genetics, Aarhus University, DK-8000 Aarhus C, Denmark; snf@mbg.au.dk; 5Laboratory of Clinical Biochemistry and Metabolism, Department of General Pediatrics, Adolescent Medicine and Neonatology, Medical Center - University of Freiburg, Faculty of Medicine, 79106 Freiburg, Germany

**Keywords:** glycogen storage disease, glycogen, vitamin B_12_, cobalamin, liver transaminases, vitamin B_12_ intake

## Abstract

*Background*: Hepatic glycogen storage diseases (GSDs) are inborn errors of metabolism affecting the synthesis or breakdown of glycogen in the liver. This study, for the first time, systematically assessed vitamin B_12_ status in a large cohort of hepatic GSD patients. *Methods*: Plasma vitamin B_12_, total plasma homocysteine (tHcy) and methylmalonic acid concentrations were measured in 44 patients with hepatic GSDs and compared to 42 healthy age- and gender-matched controls. Correlations of vitamin B_12_ status with different disease markers of GSDs (including liver transaminase activities and triglycerides) as well as the vitamin B_12_ intake were studied. *Results*: GSD patients had significantly higher plasma vitamin B_12_ concentrations than healthy controls (*p* = 0.0002). Plasma vitamin B_12_ concentration remained elevated in GSD patients irrespective of vitamin B_12_ intake. Plasma vitamin B_12_ concentrations correlated negatively with triglyceride levels, whereas no correlations were detected with liver transaminase activities (GOT and GPT) in GSD patients. Merging biomarker data of healthy controls and GSD patients showed a positive correlation between vitamin B_12_ status and liver function, which suggests complex biomarker associations. A combined analysis of biomarkers permitted a reliable clustering of healthy controls versus GSD patients. *Conclusions*: Elevated plasma concentration of vitamin B_12_ (irrespective of B_12_ intake) is a common finding in patients with hepatic GSD. The negative correlation of plasma vitamin B_12_ with triglyceride levels suggests an influence of metabolic control on the vitamin B_12_ status of GSD patients. Elevated vitamin B_12_ was not correlated with GOT and GPT in our cohort of GSD patients. Merging of data from healthy controls and GSD patients yielded positive correlations between these biomarkers. This apparent dichotomy highlights the intrinsic complexity of biomarker associations and argues against generalizations of liver disease and elevated vitamin B_12_ in blood. Further studies are needed to determine whether the identified associations are causal or coincidental, and the possible impact of chronically elevated vitamin B_12_ on GSD.

## 1. Background

Glycogen storage diseases (GSDs) are a group of inborn errors of metabolism caused by enzyme or transporter defects that disrupt the synthesis and/or breakdown of glycogen [1]. Two major clinical subtypes are distinguished: (i) hepatic GSDs that affects glycogen storage in the liver, resulting in hepatomegaly and hypoglycemia; and (ii) muscle GSDs that manifest primarily with muscle weakness and/or hypotonia. Hepatic GSDs are further classified in GSD type Ia and Ib, as well as the so-called ketotic GSDs, including types III, VI, IV and 0.

Vitamin B_12_, also called cobalamin (Cbl, B_12_), is a water-soluble B-vitamin. Humans are unable to synthesize B_12_ and thus rely on its dietary intake. In Nature, B_12_ is exclusively synthesized by a few groups of bacteria and archaea [2]. Human omnivores obtain B_12_ indirectly via the consumption of animals, which obtain vitamin B_12_ via their ruminal microbiota. Adequate intake of vitamin B_12_ in vegetarians, vegans and subgroups of patients with inherited metabolic diseases (requiring a diet free of animal products) relies on the consumption of B_12_-fortified foods or the use of oral B_12_ supplements. Vitamin B_12_ serves as a cofactor for two enzymes: cytosolic methionine synthase (MS) and mitochondrial methylmalonyl-CoA mutase (MCM). As a consequence, B_12_ deficiency results in the accumulation of unmetabolized homocysteine (Hcy) and methylmalonic acid (MMA), detectable in plasma (both) and urine (MMA). In addition to the usual analysis of total B_12_, tHcy (total plasma homocysteine) and MMA are the recommended serum biomarkers utilized to determine the overall vitamin B_12_ status [3]. 

Dietary vitamin B_12_ is carried into and through the digestive system via affinity-mediated binding to protein transporters, namely haptocorrin (HC), intrinsic factor (IF) and transcobalamin (TC) [4,5]. Free, dietary vitamin B_12_ binds first to human HC present in the upper gastrointestinal tract (saliva and gastric juice). The vitamin B_12_-HC complex undergoes proteolysis in the lower portions of the intestine wherein the micronutrient then binds to IF, which is secreted by gastric parietal cells, but binds to B_12_ only upon normalization of pH. The complex of vitamin B_12_-IF is absorbed by ileal enterocytes, and within the enterocyte IF undergoes proteolysis and B_12_ binds to the third vitamin B_12_ transporter, TC. Holo-TC enters portal circulation and is distributed to all cells in the body. Transcellular transport of vitamin B_12_ has been described for intestinal epithelial cells [6,7] and vascular endothelial cells [8]. The organ with the greatest content of vitamin B_12_ is the liver followed by kidney and spleen [4]. As a result, certain liver diseases affect vitamin B_12_ status by influencing turnover and release of the micronutrient and its protein binders from hepatocytes into circulation.

Cbl deficiency is a common condition and its underlying causes and clinical symptoms (such as neurologic deterioration and megaloblastic anemia) have been well characterized. In contrast, the causes and consequences of elevated vitamin B_12_ levels are still not fully understood. Several conditions that may result in elevated vitamin B_12_ concentrations have been described, including immunological, inflammatory, infectious, hematologic and oncologic diseases [9,10,11,12]. Severe liver diseases, such as hepatitis, hepatocellular carcinoma and cirrhosis, were also found to be associated with elevated Cbl concentrations in blood [13,14,15,16]. A study conducted with 5571 participants in the Netherlands identified that higher concentrations of plasma vitamin B_12_ (apparently reflecting some disorder) were associated with increased risk of all-cause mortality after adjusting for age, sex, and renal function among other variables [17]. In contrast to associations of elevated endogenous vitamin B_12_ with certain pathologies, exogenously administered vitamin B_12_ has virtually no toxicity, even if administered at supraphysiological concentrations as done by elite athletes [18] or to counteract cyanide poisoning [19]. 

Elevated plasma vitamin B_12_ concentrations were observed during the routine health screenings of our cohort of GSD patients in Germany. This prompted us to perform a systematic assessment of vitamin B_12_ status in patients with hepatic GSDs. We hypothesized that patients with GSD may exhibit an abnormal status of vitamin B_12_ either due to Cbl over-supplementation and/or liver pathology. The aims of this study were: (1) to assess the vitamin B_12_ status of patients with hepatic GSDs using different plasma biomarkers of vitamin B_12_ status, thereby testing for functional deficiency; (2) to elucidate potential associations between vitamin B_12_ status and liver function/metabolic control, and (3) to examine other factors (supplementation, age etc.), which might influence vitamin B_12_ levels in this patient cohort.

## 2. Patients and Methods

### 2.1. Patients

Forty-four patients with hepatic GSDs and 42 healthy age- and gender-matched controls were included in this study. The study was approved by the ethics board of the University of Freiburg (EK-Nr. 443/18). Written informed consent was obtained from all patients, patients’ parents or their legal guardians.

### 2.2. Handling of Blood Samples

All venous blood samples were drawn from the arm and collected in EDTA-tubes (Monovette EDTA/KE 9 mL, Sarstedt, Nümbrecht, Germany), centrifuged immediately at 4.900 rpm (4168× *g*), for 8 min at 4 °C and stored at −80 °C until analysis. Due to the risk of hypoglycaemia in patients with GSD, fasting time before blood draws was on average 3 h, both for GSD patients and healthy controls. To permit for more reliable comparisons, the healthy control group also included arbitrary selected individuals whose blood was collected under non-fasting conditions. 

### 2.3. Determination of Total Vitamin B_12_, Triglycerides and Transaminases in Plasma

Total vitamin B_12_ concentrations were measured using an electrochemiluminescence immunoassay (Roche, Roche Diagnostics International Ltd, Basel, Switzerland), triglycerides and transaminase activities were measured by routine techniques in the central diagnostic laboratory of the University Hospital Freiburg. Triglycerides and transaminases were analysed on a Cobas 8000 c502/C702 autoanalyser from Roche. Plasma vitamin B_12_ was analysed on a Cobas 8000 e802 autoanalyser from Roche.

GOT activity was measured at 37 °C according to the recommendations of the International Federation of Clinical Chemistry (IFCC). The GOT in the sample catalyzes the transfer of an amino group between L-aspartate and 2-oxoglutarate, producing oxaloacetate and L-glutamate. Oxaloacetate then reacts with NADH to form NAD ^+^ in the presence of malate dehydrogenase (MDH). Pyridoxal phosphate serves as a coenzyme in the amino transfer reaction, ensuring full enzyme activation. The rate of oxidation of NADH determined by decrease in absorbance at 340 nm is directly proportional to GOT activity. The linearity range is 5–700 U/L. 

GPT activity was determined at 37 °C according to the guidelines of the IFCC, in the presence of pyridoxal phosphate. GPT catalyzes the transfer of the 2-amino group from alanine to 2-oxoglutarate to form glutamate and pyruvate. Formation of product pyruvate is followed by the coupled reaction of lactate dehydrogenase whereby NADH is oxidized to form NAD^+^. The consumption of NADH is monitored by measuring the absorbance at 340 nm, which is directly proportional to the rate of pyruvate formation by GPT activity. The linearity range is 5–700 U/L. 

Serum triglycerides were determined by hydrolysis to glycerol and free fatty acids in a lipoprotein lipase-catalyzed reaction with subsequent oxidation to dihydroacetone phosphate and hydrogen peroxide. The formed hydrogen peroxide is quantified by the formation of a red dye by its reaction with 4-aminophenazone and 4-chlorophenol in the presence of peroxidase. This Trinder endpoint reaction has a linearity range of 8.85–885 mg/dL. 

Total vitamin B_12_ concentrations were measured using a competitive electrochemiluminescence immunoassay with a calibration curve from 100–2000 pg/mL. The reference range of plasma B_12_ is 198–771 pg/mL. Vitamin B_12_ concentrations above 771 pg/mL were categorized as elevated vitamin B_12_.

### 2.4. Determination of tHcy and MMA

Total plasma homocysteine concentrations were measured by tandem mass spectrometry as described earlier [20]. Briefly, 20 µL of plasma were mixed with 20 µL of DTT 0.5 M to reduce all free and protein-bound disulfides, vortexed and allowed to react at room temperature for 15 min. Twenty µL of internal standard D_4_-homocysteine (50 µM) were added and metabolites were extracted by addition of 100 µL of 0.1% formic acid in MeOH. The sample was centrifuged at 9447× *g* for 10 min at room temperature and the resulting supernatants transferred into HPLC vials for LC-MS/MS analysis as described [20]. Methylmalonic acid levels in plasma were determined using liquid chromatography- tandem mass spectrometry as described elsewhere [21]. Briefly, 50 µL of plasma were mixed with 50 µL of internal standard D3-methylmalonic acid (0.8 µM) and sample cleanup was performed by ultrafiltration in a microcentrifuge tube. The filtrate was acidified with 10 µL of 4% formic acid and the sample transferred into an HPLC vial for subsequent LC-MS/MS determination as described [21]. Both for tHcy and MMA, assay performance quality was examined by incorporating a commercially available standardized marker for plasma analysis (Control special assays in serum, product numbers SAS-02.1 and SAS-02.2, MCA Laboratories, Winterswijk, The Netherlands).

### 2.5. Calculation of the Combined Vitamin B_12_ Index (cB_12_)

The combined vitamin B_12_ index (cB_12_) was calculated as described in previous works [22,23]. The index permits a more accurate and specific assessment of vitamin B_12_ status using a combination of biomarkers. In our case, the cB_12_ index included total plasma vitamin B_12_, tHcy and MMA. The cB_12_ index was calculated using the following expression:
$$cB_{12}=\;\log_{10}{\left(\frac{holoTC\cdot B_{12}}{MMA\cdot Hcy}\right)}_{Test}\;-\;\frac{3.79}{1+{\left(\frac{age}{230}\right)}^{2.6}}\;+\;1.1\cdot e^{-\frac{folate}{3}}$$

Here the first element of equation represents the combination of four markers of B_12_-status in the test sample, the second one reflects the value expected in a reference group (with correction for age), and the last element describes correction of folate-caused shift in Hcy. In brief, the index cB_12_ describes deviation of a test sample from a “normal” reference cohort, where cB_12_ around zero (or ≥0) indicates an adequate status, whereas negative values (e.g., −1, −2, −3) describe the grades of insufficiency (e.g., low B_12_, possible deficiency, probable deficiency).

### 2.6. Estimation of Vitamin B_12_ Intake

Oral vitamin B_12_ intake and possible additional supplementation were assessed using a questionnaire that addressed the patients’ nutritional habits. The questionnaire specifically focused on the frequency of consumption of vitamin B_12_-containing foods such as different types of meat, fish, eggs and dairy products, as well as on the supplementation with vitamin preparations containing vitamin B_12_. Vitamin B_12_ intake was estimated by multiplying the frequency of consumption by the average vitamin B_12_ content of these foods as given in the nutritional table published by the German Society of Nutrition (DGE) [24]. The questionnaire, evaluating Cbl intake, is provided in Appendix A. 

### 2.7. Statistical Analyses

Statistical analyses and data fitting were performed using GraphPad Prism 7 (GraphPad Software, Inc., La Jolla, CA, USA) and KyPlot 5 (freeware available from KyensLab Inc., Tokyo, Japan). Normal distribution of the data was assessed with the D’Agostino & Pearson test and the Shapiro-Wilk test. Statistical differences were calculated by the Mann-Whitney-U-Tests for non-parametrically distributed data. Correlations were tested using the Spearman’s rank correlation coefficient and Theil-Sen estimator (for prognosis of a robust linear connection, insensitive to outliers). Because the data for some biomarkers were markedly skewed (Appendix A) we examined correlations after logarithmic transformation of the data. The significance level was set to α = 0.05.

## 3. Results

### 3.1. Characteristics of the Cohort

Forty-four patients with hepatic GSDs and 42 healthy age- and gender-matched controls from different regions of Germany were enrolled in this study. Biometric and biochemical parameters of the healthy control- and GSD-groups are provided in Table 1 and Appendix A. The representation of GSD-subtypes in the GSD-group is summarized in Figure 1. 

The mean age of the patient group was 20 years (range 2–59 years), the mean age of the control group was 22 years (range 1–62 years). 50% of GSD patients and 52% healthy controls were over 18 years, respectively. 20 patients were female, 24 were male. The reference control group consisted of 19 females and 23 males. 

### 3.2. Assessment of Vitamin B_12_ Intake and Correlation between Plasma Vitamin B_12_ Concentration and Vitamin B_12_ Intake

An estimation of the daily vitamin B_12_ intake was accomplished for 32 patients who completed the questionnaire with all necessary details. The intake varied between patients and ranged from 1.56 µg/day (1.15 nmol/day, a low-normal dose) to 1007 µg/day (≈1400 nmol/day, a high dose). The latter case was registered in one patient who ingested vitamin B_12_ at a quantity of 1000 µg/day). In four patients (12.5%), the daily intake was lower than the recommended 1.5 µg/day to 4.0 µg/day, depending on age, according to the German Society of Nutrition (DGE e.V.) [25]. Three patients had an intake exceeding the recommended daily dose by a factor of 3 or more. Two of the latter three patients (all with GSD I), had elevated vitamin B_12_ levels in plasma. While previous studies have shown that the intake of vitamin B_12_ correlates with plasma concentration of the micronutrient in healthy subjects [26,27], no overall correlation was observed between plasma vitamin B_12_ concentration and the daily vitamin B_12_ intake in GSD patients (Appendix A, *p* = 0.36). 

### 3.3. Plasma Concentration of tHcy, MMA and Vitamin B_12_

Total plasma concentrations of vitamin B_12_, tHcy and MMA of GSD patients and healthy controls are shown in Figure 2 and Table 1. Plasma concentrations of vitamin B_12_ were significantly higher in GSD patients compared to controls (mean values of 667 pg/mL and 379 pg/mL, respectively, *p* = 0.0002, Table 1). In contrast, no significant differences in the concentrations of tHcy and MMA were observed (*p* = 0.54 and *p* = 0.71, respectively, Table 1). Elevated vitamin B_12_ concentrations were found in 13/44 (29.5%) of patients and only 3/42 (7.1%) of controls. Vitamin B_12_ deficiency was stipulated as a plasma B_12_ concentration < 198 pg/mL, and it was detected in 1 of the 44 patients (2.3%) and none of the controls. Elevated vitamin B_12_ levels were more common in patients with ketotic GSDs compared to GSD I (50% versus 22%, respectively); however, this difference did not reach statistical significance (*p* = 0.069). Elevated MMA concentrations were found in 4/44 of patients (9.1%) and 4/40 (10%) of controls (*p* = 0.71). The concentration of tHcy was within the normal reference range both in healthy controls and GSD patients. 

### 3.4. Combined Vitamin B_12_ Index, cB_12_

The reliable assessment of vitamin B_12_ status requires the measurement of two to four biomarkers [3,22,23,28,29]. Herein, tHcy, MMA and vitamin B_12_ values were combined into the cB_12_ index to more accurately examine the status of vitamin B_12_. The results of such assessment are provided in Table 2 and show that 18.2% of GSD patients and 2.4% of controls were identified with elevated vitamin B_12_, while only 2 patients (4.5%) were categorized as having decreased vitamin B_12_. Thus, the vast majority of patients in our cohort exhibited an adequate vitamin B_12_ status. Noteworthy, a comparison of vitamin B_12_ concentration of healthy controls versus GSD patients within the group ranked as having an adequate vitamin B_12_ status showed that GSD patients exhibit higher concentration of vitamin B_12_ compared to healthy controls (median healthy controls: 248 pg/mL; median GSD patients: 360 pg/mL, *p* = 0.001, Mann-Whitney test). 

### 3.5. Associations between Vitamin B_12_ Status and Age, Gender and BMI in GSD

Certain biomarkers of vitamin B_12_ status vary with age, gender and, possibly, BMI in the healthy population [30,31,32,33]. Here, we examined whether such associations hold true for both the GSD patients and our healthy control group. Results from correlation analysis are shown in Table 3 and Table 4, respectively. High vitamin B_12_ concentrations were associated with low levels of tHcy, MMA and triglycerides in the GSD patient group, while no correlation was detected with age or BMI. 

### 3.6. Associations between Vitamin B_12_ Status and Triglycerides and Liver Transaminases

Correlation between plasma vitamin B_12_, triglycerides and liver enzymes GOT and GPT based on logarithmic analysis of the data, which suppressed excessive dispersion of the data, are presented in Figure 3 and Appendix A. 

For comparative purposes, results from statistical analysis on non-transformed datasets are given in Table 3 and Table 4 as well. As expected, GSD patients had significantly higher triglyceride concentrations than healthy controls (*p* < 0.0001, Table 1). Some associations have been described between vitamin B_12_ status and triglyceride/lipid metabolism in human and animal studies [34,35,36,37,38]. Thus, we investigated whether associations exist between vitamin B_12_ and triglycerides concentrations in healthy controls and GSD patients. No significant correlation was identified between plasma vitamin B_12_ and triglycerides in the control group (*p* = 0.46), but a negative correlation was observed between these biomarkers in the GSD group (*p* = 0.008) (Figure 3, panels A and B, respectively, and also Appendix A and Table 3). GSD patients had significantly higher GOT and GPT levels than controls (*p* < 0.0001 for GOT, *p* < 0.0001 for GPT, Table 1). Early and current literature have hypothesized that one of the reasons for unexplained elevated vitamin B_12_ in plasma is liver dysfunction [13,16,39,40,41,42]. We sought out to examine whether liver injury seen in GSD patients is associated with changes in plasma vitamin B_12_ status. The results of correlations between vitamin B_12_ concentrations and liver transaminases are shown in Figure 3, panels C to F (and see also Table 4 and Appendix A for results on non-log10 transformed data). No correlations were found between log10 vitamin B_12_ concentration and log10 GOT or GPT in the GSD group though in all cases a positive upward drift was noticed. A positive correlation between log10 vitamin B_12_ and log10 GOT was seen only for the healthy control group (*p* = <0.001).

### 3.7. Relationships between Biomarkers upon Merging Healthy Control and GSD Groups as a Continuum

To further study the dependencies of the variables chosen in this study, we merged data from healthy controls and GSD patients. The motivation for this analysis was two-fold. First, we searched for a better diagnostic separation of the two groups under study (e.g., via a two-dimensional presentation of B_12_ vs an established GSD-marker). Secondly, we attempted to define the vector of metabolic transformation “healthy → GSD” within boundaries of two GSD-markers Y and X (including B_12_). In all cases, we aimed to test possible usability of B_12_ in GSD-diagnostics. Correlations between Y and X were examined by the Theil-Sen method, which estimates median slope and has a low sensitivity to outliers. The X,Y- and Y,X-fitting lines were calculated and used as a frame to draw the overall mean vector “healthy → GSD” together with a predicted 2D cut-off line, which produced the best separation between controls (red points) and GSD-patients (green points). The correlation between B_12_ and TG (Figure 4, panel A) was not statistically significant, yet the probability of equal medians for the cohorts of controls and patients was very low: *p* = 0.0002 (along X-axis) and *p* = 10^−8^ (along Y-axis), pointing to an association of the two variables. The approximate cut-off lines (yellow arrows in Figure 4) were drawn and gave reasonably low scores of the “misplaced points” (reds among greens/greens among reds): (7/4), (11/11), (10/11), (6/6) for panels A, B, C, D, respectively. Strictly vertical or horizontal cut-offs (based on a single variable, either X or Y) revealed greater overlaps. For example, panel D gave the counts (11/12) for the X-based cut off and (7/8) for the Y-based cut-off, both inferior to (6/6) obtained by 2D method with an angled separator (yellow double arrow). We next examined associations between vitamin B_12_ and distinct mathematical combinations of biomarkers relevant to GSD. Such combined analysis of biomarkers reduces the contribution of their individual variabilities, thus permitting a more robust discrimination between healthy and diseased subjects, as discussed in another context in refs. [22,23,43]). Figure 5 explores logarithmic dependencies of B_12_ plotted versus several combinations of GSD-markers. A statistically significant correlation of the combined GSD-markers and B_12_ was identified in all cases for ‘healthy → GSD’ vectors (see *p* values in Figure 5), albeit not as strong as the correlation seen between B_12_ versus GOT (Figure 4, panel B, the lowest *p*, biomarkers not combined). Yet the panels in Figure 5 indicated a better separation of points for the cohorts of healthy controls and GSD patients. Thus, the counts of “misplaced” points (reds among greens/greens among reds) were assessed as (8/8), (6/5), (6/5) in panels A, B, C, respectively. These values are generally lower than those in Figure 4, where only two variables (one for each axis) were used. Probabilities of overlapping distributions were further assessed in a unidimensional space (with a simple or combined variable X), and the results are presented in Table 5. Optimal dissection of the two cohorts was obtained for ½·log10(GPT·TG), followed by ¼·log10(B12·GOT·GPT·TG) and ⅓·log10(GOT·GPT·TG), according to the lowest probabilities of overlap. None of the individual markers came close to the effectiveness of the combined markers in discriminating the profiles of healthy controls and GSD patients.

## 4. Discussion

We present the first systematic assessment of vitamin B_12_ status in a large cohort of hepatic GSD patients compared to healthy age- and gender-matched controls. This work was initiated upon the observation of abnormally elevated plasma vitamin B_12_ concentrations in our cohort of GSD patients. The major findings of this study are: (1) Plasma vitamin B_12_ concentration is significantly higher in our study cohort (44 hepatic GSD patients) compared to healthy controls; (2) The elevation of plasma vitamin B_12_ concentrations is not explained by dietary or medical over-supplementation with Cbl; (3) Elevated plasma vitamin B_12_ does not associate with liver function biomarkers GOT and GPT in our cohort of patients with GSD, even though the overall vector ‘healthy → GSD’ showed a correlation upon a hypothetical ‘jump’ from one metabolic state to another; (4) Vitamin B_12_ levels seem to correlate with metabolic control in hepatic GSD patients (negative correlation of B_12_ versus TG); (5) None of the GSD patients showed functional vitamin B_12_ deficiency; and (6) Combined assessment of GSD markers plus B_12_ helps to discriminate between healthy controls and GSD cases.

### 4.1. Causes and Biological Activity of Elevated Plasma Vitamin B_12_

In contrast to vitamin B_12_ deficiency, the pathophysiology and clinical consequences of high serum cobalamin have been insufficiently studied. Possible mechanisms for increased vitamin B_12_ concentrations in plasma comprise excess intake or administration of Cbl, hepatic release of Cbl (and its slow-exchanging blood transporter HC), an increased secretion of HC from malignant tissues, and an increase in the fast exchanging transporter TC via excess production or lack of clearance. In most cases studied so far, elevation of plasma vitamin B_12_ was caused by elevation of its transport protein HC, which has a very slow turnover and can retain considerable quantities of B_12_ in blood [44,45]. Plasma transporters of vitamin B_12_ (holo-TC and HC) and unsaturated cobalamin binding capacity (UCBC) [46] were not determined in this study. Therefore, it is currently unknown whether GSD patients with highly elevated vitamin B_12_ have a saturated cobalamin binding capacity of its authentic transport proteins. If so, excess vitamin B_12_ might bind to non-dedicated proteins such as albumin [47,48,49,50] and the recently described immunocomplexes [51,52,53]. The biological activity of these protein complexes of vitamin B_12_ is unknown and currently under investigation. 

### 4.2. Biomarkers of Vitamin B_12_ Status in GSD

Vitamin B_12_ biomarkers are influenced by diet, supplementation with vitamin B_12_ and certain medications [3]. Unsupplemented patients who must adhere to special diets with a reduced content of meat, dairy products and eggs are at risk of developing vitamin B_12_ deficiency. This has been well described for inherited disorders of amino acid metabolism, such as phenylketonuria [54,55]. The recommended diet in GSD I is rich in carbohydrates, resulting in a lower intake of protein and fat [56]. It can therefore be assumed that patients with GSD I are at risk of vitamin B_12_ deficiency. Yet, we identified by the combined vitamin B_12_ index only two GSD Ia patients as having a decreased vitamin B_12_ status, while the rest were adequate in this regard. The mainstay of therapy in ketotic GSDs is a protein-rich diet. It usually contains high amounts of meat and dairy products, which would provide adequate vitamin B_12_ intake, thus making a deficiency unlikely in these cases. Although we did not find a correlation between the estimated vitamin B_12_ intake and plasma vitamin B_12_ concentrations in our study cohort, elevated vitamin B_12_ levels were more common in ketotic GSD patients compared to GSD I patients (50% vs. 22%). This could be due to a distinct effect of GSD on plasma vitamin B_12_ concentration or due to statistical bias introduced by the small size of the cohort examined herein. Besides, we acknowledge that the estimated vitamin B_12_ intake determined via our nutrition questionnaire only allows for an educated guess of this variable as opposed to the accuracy that is achieved via standardized dietary intake protocols. Other studies performed with special populations such as elders with abnormal vitamin B_12_ biomarkers [57] or with atrophic gastritis [58] also showed a lack of correlation between intake and plasma concentration of vitamin B_12_. Therefore, the relationships between intake of vitamin B_12_ and its plasma concentrations in disease merits further investigation. 

One limitation of this study is that vitamin B_12_ intake was only assessed in the GSD patient group and not in the control group. Nonetheless, results of the measurement of plasma vitamin B_12_, tHcy and MMA strongly suggest that most subjects of our study had an adequate vitamin B_12_ intake. A number of studies point to the lack of correlation between serum and cellular levels of vitamin B_12_ [59,60,61]. Normal B_12_ concentrations in plasma or serum do not exclude B_12_ deficiency at the cellular level. The combination of normal or even elevated plasma B_12_ levels with simultaneously elevated concentration of MMA and/or tHcy is called functional vitamin B_12_ deficiency. For other metabolic diseases it has been shown that some patients develop vitamin B_12_ deficiency despite supplementation of vitamin B_12_ at a recommended daily dose [54]. In a study by Vugteveen et al. 2011 with 75 phenylketonuria patients, 10 patients with normal vitamin B_12_ concentrations in serum showed elevated plasma concentrations of MMA and/or tHcy [54]. In contrast to these data, none of our GSD patients had a functional vitamin B_12_ deficiency. 

### 4.3. Vitamin B_12_ Status and Liver Disease in GSD

Elevated vitamin B_12_ concentrations have been reported as markers for liver pathologies and liver damage such as cirrhosis, hepatitis or hepatocellular carcinoma [13,16,39,40,41,42,45]. Although many of the GSD patients in our cohort had elevated liver transaminase activities, the assessment of associations between vitamin B_12_ status and liver transaminase activities using Spearman’s linear correlation did not retrieve statistical significance. Liver pathologies are complex. The form of liver damage observed in cirrhosis, infection, or a metabolic disease like GSD are not comparable physiologically. Therefore, associations between elevated vitamin B_12_ and liver function previously described in other hepatic diseases cannot be superimposed onto liver damage caused by GSD. From the mathematical point of view, the power of the statistical assessment in the current work is limited by the small number of subjects available for study in the field of rare diseases. Thus, the number of patients in our study amounted to 44 individuals with hepatic GSD. The naturally skewed profile of the biological parameters examined was partially compensated by the logarithmic transformation, yet some bias might have been added anyway. Keeping in mind all the aforementioned issues, we report here a negative correlation between vitamin B_12_ concentrations and triglycerides. In studies with GSD I patients, triglycerides were shown to be a better marker for metabolic control than parameters such as lactate, glucose, liver transaminases or uric acid [62], and triglyceride levels below 200 mg/dl are one criterion for optimal metabolic control in GSD I [62]. The reduced tolerance to fasting made it impossible to collect blood and determine triglyceride concentrations under conditions of fasting for some of the GSD patients. Despite this caveat, the correlation between plasma vitamin B_12_ and triglyceride levels found in our study suggests an association between vitamin B_12_ status and metabolic control. Studies performed in rats subjected to a vitamin B_12_ restricted diet showed elevation of triglycerides that was transmissible to the offspring [63]. Follow-up work revealed that vitamin B_12_ deficiency altered DNA methylation in rats including genes involved in fatty acid metabolism [64]. The authors suggested that the association between vitamin B_12_ status and triglycerides is causal, as rehabilitation of the vitamin B_12_-restricted animals with a diet rich in vitamin B_12_ corrected the abnormal concentrations of triglycerides [63]. Further studies are needed to investigate whether the association between plasma vitamin B_12_ and triglycerides identified in our study is causal or coincidental. 

### 4.4. Behaviour of Biomarkers in a Mathematical Continuum of Healthy Controls and GSD Patients

The associations between biomarkers were also investigated when the data from healthy controls and GSD patients were merged together. This analysis ignores the fact that GSD patients carry a gene mutation that modifies their biomarkers of metabolism since birth, making them a population medically and irreversibly distinct form healthy controls. Instead, we herein focused on how the biomarkers themselves behave in a theoretical continuum between healthy controls and GSD patients. Our analysis showed positive associations between plasma vitamin B_12_ and liver function markers GOT and GPT. The analysis also showed significantly different median values of B_12_ and TG, despite the lack of correlation between these biomarkers. The apparent dichotomy of intra-cohort and merged-cohort analyses suggests that the associations between these biomarkers are complex and cannot be generalized via straightforward statements like ‘all liver diseases lead to an elevated plasma vitamin B_12′_ or ‘absent correlation between markers X,Y within the groups A and B precludes discrimination of A and B via X-test and/or Y-test’. 

GSDs disrupt carbohydrate metabolism thereby impairing the major source of energy in the cell. Vitamin B_12_ is an important element of cellular energy metabolism supporting the biosynthesis of methionine and succinyl-CoA. A recent epidemiological study identified heritability of the combined vitamin B_12_ index and previously unrecognized associations of vitamin B_12_ status with mitochondrial substrates and energy metabolism [65]. Further work is ongoing in our group to elucidate possible mechanistic implications of vitamin B_12_ status in energy metabolism in patients with GSD and in other defects in energy-producing pathways. 

### 4.5. Clinical Implications of Elevated Vitamin B_12_ in Patients with Hepatic GSD

Elevated vitamin B_12_ concentrations are a common finding in hepatic GSD patients that may represent a previously unrecognized hallmark of this disease. Similar to elevated biotinidase activity found in most patients with hepatic GSD [66] plasma vitamin B_12_ may harbor diagnostic value. A combination of B_12_ measurements with other markers might improve the diagnosis reliability. For instance, Table 5 shows that no single marker can separate healthy controls from GSD patients as effectively as is achieved with a combination of several markers (if working in a unidimensional space). Comparison of Figure 4 versus Figure 5 illustrates the same finding in a two-dimensional space (particularly visible for Panel 4B versus Panel 5C). The underlying mechanisms as well as the possible impact of metabolic control on plasma vitamin B_12_ concentration warrants further investigation. Long-term monitoring of vitamin B_12_ concentrations in the clinical course of patients may help to elucidate influencing factors, such as diet, metabolic control or medication.

## 5. Conclusions

Our study confirmed that elevated plasma concentrations of vitamin B_12_ in patients with hepatic GSD are a common finding that is not explicable by a high vitamin B_12_ intake or over-supplementation of Cbl. Elucidating the fate, biological activity and health outcomes of elevated plasma vitamin B_12_ in humans remains a matter of active investigation. While no correlation between vitamin B_12_ status and liver transaminase activities was found in the GSD cohort, a negative correlation of plasma vitamin B_12_ with triglyceride levels hints to a possible impact of metabolic control on the vitamin B_12_ status of GSD patients. Further studies are required to determine the causal or coincidental nature of the associations identified, and the possible impact of chronically elevated vitamin B_12_ on GSD pathogenesis and outcomes. Finally, the joint analysis of B_12_, TG and liver function markers GOT and GPT showed distinct clustering of the data, which provided a better separation of healthy controls from GSD patients. This supports the notion that examining the metabolic landscape of patients via the combined contribution of biomarkers is superior compared to the analysis of the individual tests.

## Figures and Tables

**Figure 1 jcm-09-02326-f001:**
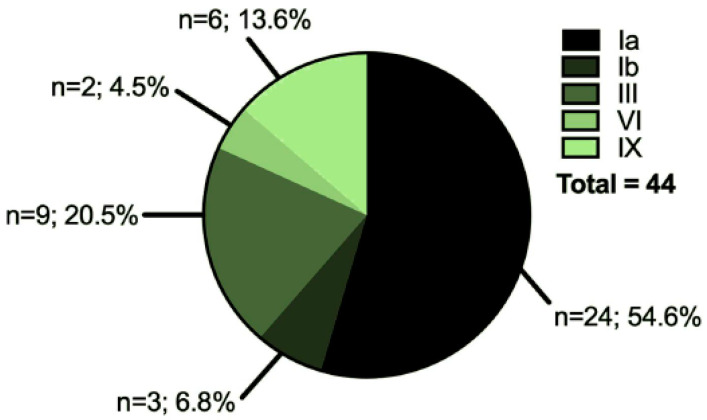
Composition of the GSD patient cohort. The study comprised a cohort of forty-four patients aged 2–59 years old from all over Germany. Twenty of the patients were female and twenty-four were male. 62% of patients had GSD I (Ia and Ib), while the remaining 38% had a ketotic form of hepatic GSD.

**Figure 2 jcm-09-02326-f002:**
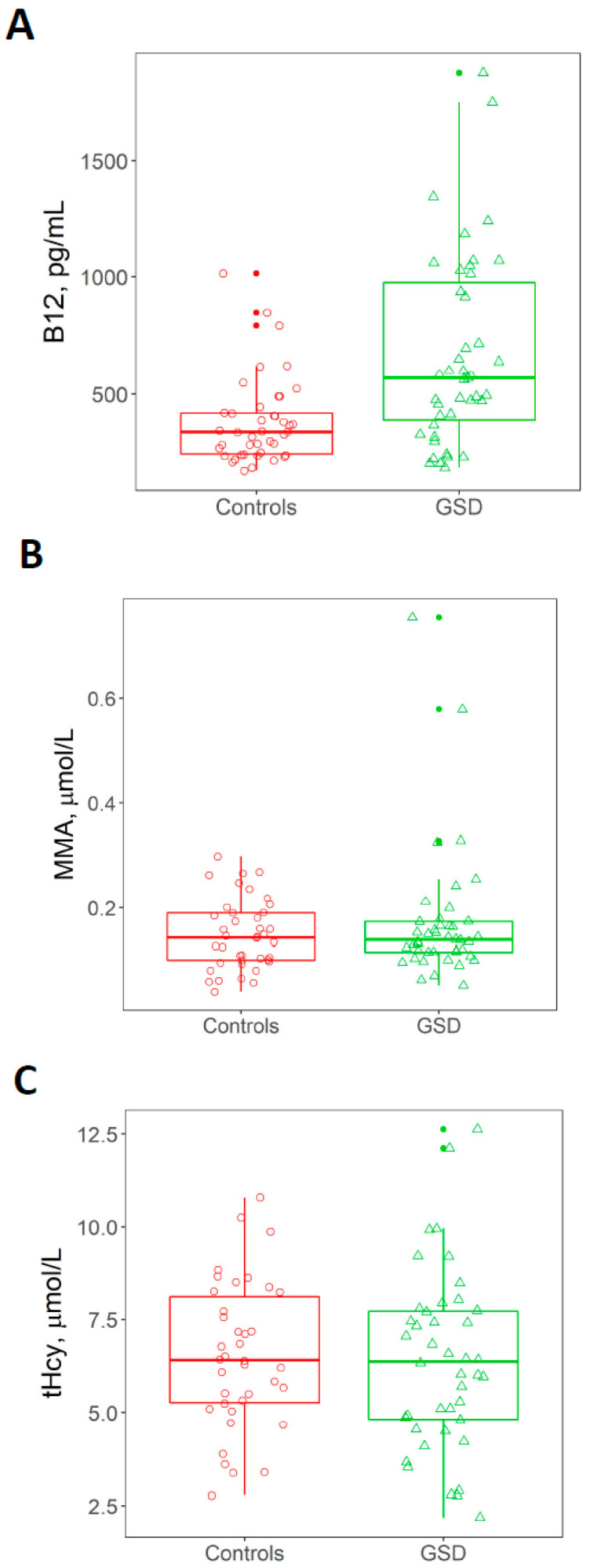
Biomarkers of vitamin B_12_ status in healthy controls and GSD patients. Panel (**A**) Plasma vitamin B_12_; Panel (**B**) Plasma MMA; Panel (**C**) Plasma tHcy. Median vitamin B_12_ concentrations were significantly higher in GSD patients compared to healthy controls (*, *p* = 0.0002), whereas no statistically significant differences were found for homocysteine and MMA between the control and GSD patients. The middle line represents the 50th quartile (median). The lower and upper hinges represent the 25th and 75th quartiles, respectively. The distance between the 25th and the 75th quartiles depicts the inter-quartile range (IQR). The upper whisker represents the largest value that extends no further than 1.5 times the IQR. The lower whisker represents the smallest value that extends up to 1.5 times the IQR.

**Figure 3 jcm-09-02326-f003:**
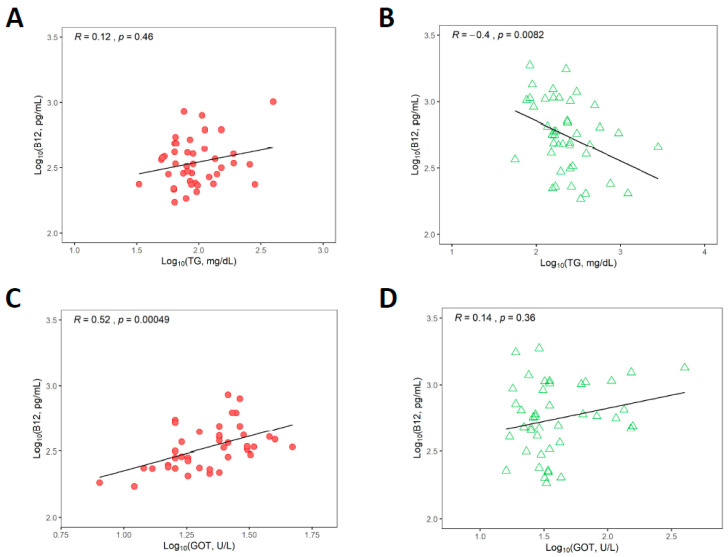
Correlation analysis (Spearman method) of log10-transformed biomarker datasets. Panel (**A**) B_12_ and TG in healthy controls; Panel (**B**) B_12_ and TG in GSD patients; Panel (**C**) B_12_ and GOT in healthy controls; Panel (**D**) B_12_ and GOT in GSD patients; Panel (**E**) B_12_ and GPT in healthy controls and Panel (**F**) B_12_ and GPT in GSD patients. Rho values for these correlations are provided in Table 5. Statistically significant associations were found for log10 B_12_ versus log10 TG in GSD patients, and for log10 B_12_ versus log10 GOT in the healthy control group.

**Figure 4 jcm-09-02326-f004:**
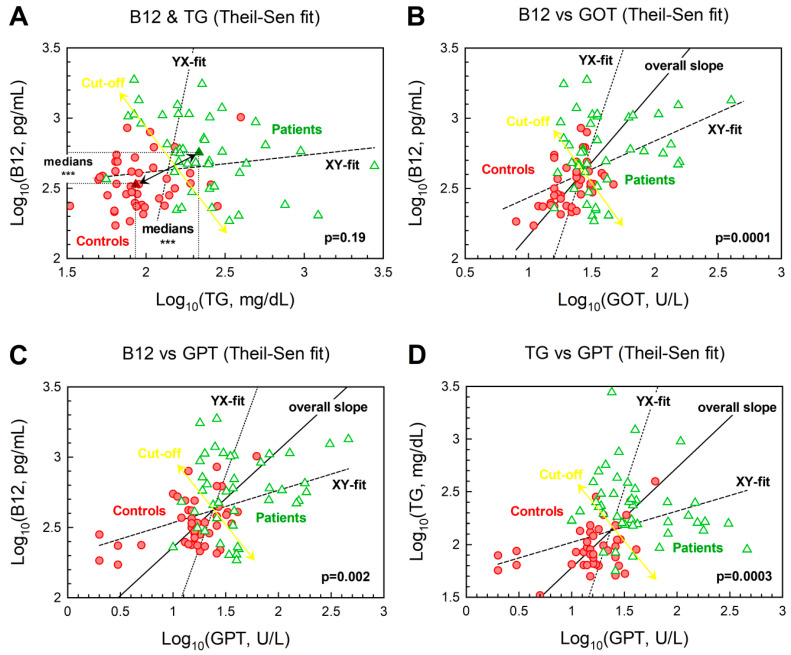
Correlations (Theil-Sen method) between biomarkers presented in logarithmic coordinates. Panels show (**A**) Triglycerides (TG) and vitamin B12; (**B**) GOT and B_12_; (**C**) GPT and B_12_; (**D**) GPT and TG. Data from cohorts of controls and GSD patients are notated as red circles and green triangles, respectively. Linear fitting was done by the median-based Theil-Sen method, and the fitting functions correspond to the direct X,Y-coordinates (dashed line, y = b0x + b1x·x) and the inverted Y,X-coordinates (dotted line, x = b0y + b1y·y). The overall slope vector of transformation of ‘healthy →GSD’ is shown as a solid line (for the cases, where the probability of zero slope is below 0.05). Cut-offs between 2D distributions of controls and GSD are indicated as yellow double-headed arrows. Panel A also shows medians of the two cohorts, as well as probabilities of equal sets (controls = patients), assessing them along axes X and Y.

**Figure 5 jcm-09-02326-f005:**
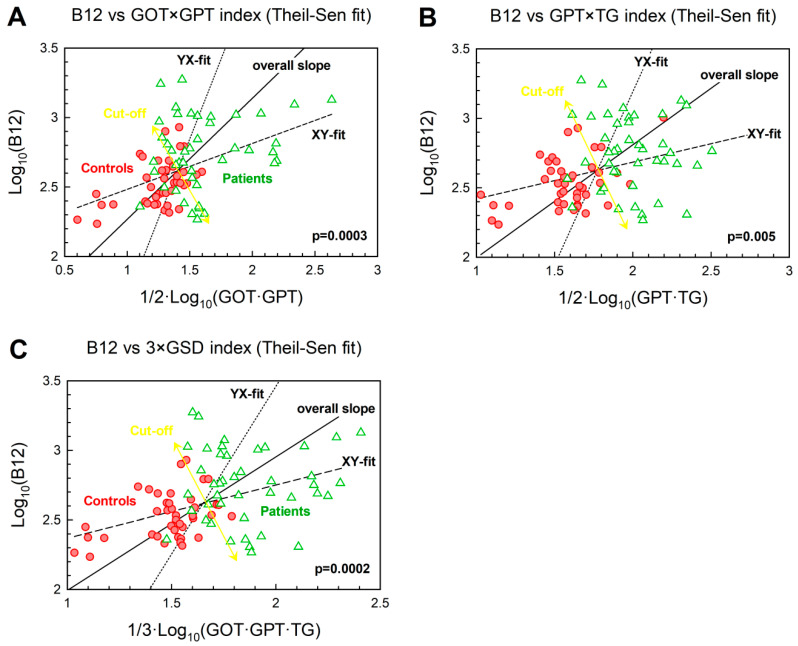
Correlations between the combined markers of GSD and B_12_ presented in logarithmic coordinates. Panels show (**A**) B_12_ vs. GOT & GPT; (**B**) B_12_ vs. GOT & TG; (**C**) B_12_ vs. GOT, GPT & TG. Data from cohorts of controls and GSD patients are notated as red circles and green triangles, respectively. All other notations are as in legend to Figure 4.

**Table 1 jcm-09-02326-t001:** Comparison of biometric and biochemical parameters determined in this study.

	Healthy Controls	GSD Patients	Healthy Controls vs. GSD Patients
**Biometric parameters and vitamin B_12_ intake**			
Age	22.67 ± 14.67 (1–62)	20.77 ± 13.39 (2–59)	NA
BMI	22.22 ± 5.37 (13.7–38.4)	22.54 ± 4.82 (13.7–35.6)	NA
Gender	19 females, 23 males	20 females, 24 males	NA
Vitamin B_12_ intake ** (µg/day)	ND	36.59 ± 177.20 (1.56–1007.47)	NA
**Biochemical parameters determined in plasma**			
Triglycerides (TG) (mg/dL)	107.64 ± 69.44 (33–397)	387.32 ± 573.89 (56–2780)	<0.0001 *
MMA (µmol/mL)	0.149 ± 0.07 (0.043–0.2985)	0.172 ± 0.124 (0.054–0.751)	0.7132 *
tHcy (µmol/L)	6.53 ± 1.96 (2.77–10.79)	6.32 ± 2.42 (2.18–12.62)	0.5452 *
Vitamin B_12_ (pg/mL)	379 ± 182.93 (172–1015)	667.28 ± 408.83 (185–1876)	0.0002 *
GOT (ASAT) (IU/L)	23.02 ± 8.26 (8–47)	56.86 ± 66.09 (16–401)	<0.0001 *
GPT (ALAT) (IU/L)	20.03 ± 10.89 (<2.5–62)	66.23 ± 84.94 (10–461)	<0.0001 *

* Comparisons between healthy controls and GSD patients were examined by Mann Whitney test; ** Daily vitamin B_12_ intake was calculated using a food assessment questionnaire (Appendix A); ND: not determined; NA: not applicable.

**Table 2 jcm-09-02326-t002:** Combined vitamin B12 index in GSD patients and controls.

	Adequate Vitamin B_12_ Status	Increased Vitamin B_12_ Status	Decreased Vitamin B_12_ Status
Patients (*n* = 44)	34 (77.3%)	8 (18.2%)	2 (4.6%)
Controls (*n* = 42)	41 (97.6%)	1 (2.4%)	0 (0%)

**Table 3 jcm-09-02326-t003:** Correlation analyses in healthy controls. Correlations were tested using the Spearman’s rank correlation coefficient. The significance level was set to α = 0.05.

Selected Variables:	Correlation Coefficient	95% Confidence Interval	*p* Value
Vitamin B_12_ and age	−0.11	−0.41 to 0.21	0.475
Vitamin B_12_ and BMI	−0.11	−0.42 to 0.22	0.516
Vitamin B_12_ and MMA	0.12	−0.20 to 0.43	0.442
Vitamin B_12_ and tHcy	−0.22	−0.51 to 0.11	0.181
Vitamin B_12_ and GOT (ASAT)	0.52	0.24 to 0.72	<0.001
Vitamin B_12_ and GPT (ALAT)	0.19	−0.14 to 0.48	0.243
Vitamin B_12_ and triglycerides	0.12	−0.20 to 0.41	0.46
MMA und tHcy	0.26	−0.15 to 0.50	0.261
MMA and triglycerides	0.38	0.06 to 0.62	0.016
tHcy and triglycerides	0.21	−0.13to 0.50	0.206
MMA and BMI	0.17	−0.17 to 0.47	0.319
tHcy and BMI	0.51	0.20 to 0.72	0.002
MMA and age	0.53	0.25 to 0.73	<0.001
tHcy and age	0.43	0.12 to 0.66	0.007

MMA: methylmalonic acid, tHcy: total homocysteine, BMI: body mass index.

**Table 4 jcm-09-02326-t004:** Correlation analyses in GSD patients. Correlations were tested using the Spearman’s rank correlation coefficient. The significance level was set to α = 0.05.

Selected Variables:	Correlation Coefficient	95% Confidence Interval	*p* Value
Vitamin B_12_ and age	−0.17	−0.45 to 0.15	0.275
Vitamin B_12_ and BMI	−0.24	−0.51 to 0.07	0.114
Vitamin B_12_ and MMA	−0.43	−0.65 to −0.14	0.004
Vitamin B_12_ and tHcy	−0.48	−0.69 to −0.20	0.001
Vitamin B_12_ and GOT (ASAT)	0.14	−0.17 to 0.43	0.356
Vitamin B_12_ and GPT (ALAT)	0.12	−0.19 to 0.41	0.435
Vitamin B_12_ and triglycerides	−0.40	−0.63 to −0.10	0.008
Vitamin B_12_ and vitamin B_12_ intake	0.24	−0.13 to 0.56	0.193
MMA and tHcy	0.11	−0.20 to 0.41	0.474
MMA and triglycerides	−0.08	−0.37 to 0.23	0.606
tHcy and triglycerides	0.39	0.09 to 0.63	0.01
MMA and BMI	−0.17	−0.45 to 0.14	0.278
tHcy and BMI	0.47	0.19 to 0.68	0.001
MMA and age	−0.12	−0.40 to 0.19	0.443
tHcy and age	0.52	0.25 to 0.71	<0.001

MMA: methylmalonic acid, tHcy: total plasma homocysteine, BMI: body mass index.

**Table 5 jcm-09-02326-t005:** Effectiveness of single and combined markers in discriminating controls from GSD patients.

Single Marker	Probability * Controls = Patients	Combined Markers	Probability * Controls = Patients
total B12	2 × 10^−4^	½ · log_10_(B_12_·TG)	4 × 10^−11^
GOT	4 × 10^−6^	½ · log_10_(GOT·GPT)	2 × 10^−7^
GPT	2 × 10^−7^	⅓ · log_10_(GOT·GPT·TG)	4 × 10^−12^
TG	1 × 10^−8^	¼ · log_10_(B_12_·GOT·GPT·TG)	3 × 10^−12^
		½ · log_10_(GPT·TG)	2 × 10^−15^

* Wilcoxon (Mann-Whitney) non-parametric test.

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
