# Peer review of "Elevated Plasma Vitamin B12 in Patients with Hepatic Glycogen Storage Diseases"

_jcm, 2020, doi:10.3390/jcm9082326_

Round 1
Reviewer 1 Report
In the present paper, Hinkel and colleagues presented the results of vitamin B12 assessment in GSD patients and in matched healthy controls and provided results of the correlations between vitamin B12 status and selected biochemical variables. The authors present novel data and statistical analysis is robust.
- In results (3.3), how are "elevated vitamin B12 concentrations" defined?
- In 3.3., can the authors provide mean comparisons of vit. B12 concentrations among groups, instead/in addition to contingency tables on the rate of elevated B12 levels between groups?
- Figure 2. Please add p values, or at least asterisks where appropriate.
- Results, 3.4. Are there any significant differences in the proportion of "adequate B12 status" between patients and controls?
- Table 3 and 4. I suggest using 2 or maximum 3 significant digits for correlation coefficients and CIs instead of 4.
- While the associations between vitamin B12 status and liver function tests are adequately described, the discussion does not convey the clinical utility of the authors' findings. Why clinicians working in the field of GSDs should pay attention to elevated vitamin B12 levels? Are they associated with the degree of liver impairment? How these markers can potentially improve the assessment of these patients?
Author Response
Comments and Suggestions for Authors
In the present paper, Hinkel and colleagues presented the results of vitamin B12 assessment in GSD patients and in matched healthy controls and provided results of the correlations between vitamin B12 status and selected biochemical variables. The authors present novel data and statistical analysis is robust.
- In results (3.3), how are "elevated vitamin B12 concentrations" defined?
Response: Thanks for noting this oversight. Elevated vitamin B12 was defined as a serum vitamin B12 concentration above 771 pg/mL. This is now indicated in Methods.
- In 3.3., can the authors provide mean comparisons of vit. B12 concentrations among groups, instead/in addition to contingency tables on the rate of elevated B12 levels between groups?
Response: We are sorry for the oversight. In section 3.3, we failed to refer to this precise comparison, which was provided in Table 1. Table 1 lists mean values for vitamin B12 concentration in each study group, and the p value from the comparison among groups. We have now updated section 3.3. to guide the reader into this result, as follows:
Total plasma concentrations of vitamin B12, tHcy and MMA of GSD patients and healthy controls are shown in Figure 2 and Table 1. Plasma concentrations of vitamin B12 were significantly higher in GSD patients compared to controls (mean values of 667 pg/ml and 379 pg/ml, respectively, p=0.0002, Table 1). In contrast, no significant differences in the concentrations of tHcy and MMA were observed (p=0.54 and p=0.71, respectively, Table 1).
Figure 2. Please add p values, or at least asterisks where appropriate.
Response: Done. Asterisk added and p value listed in the corresponding Figure legend.
- Results, 3.4. Are there any significant differences in the proportion of "adequate B12 status" between patients and controls?
Response: Thank you for this important question! Yes, vitamin B12 concentrations in healthy controls differed from those in GSD even when both groups classify as having adequate vitamin B12 status (p = 0.001). This result is now added in section 3.4, as follows:
Noteworthy, a comparison of vitamin B12 concentration of healthy controls versus GSD patients within the group ranked as having an adequate vitamin B12 status showed that GSD patients exhibit higher concentration of vitamin B12 compared to healthy controls (Median healthy controls: 248 pg/mL; Median GSD patients: 360 pg/mL, p = 0.001, Mann-Whitney test).
- Table 3 and 4. I suggest using 2 or maximum 3 significant digits for correlation coefficients and CIs instead of 4.
Response: Thanks, these have now been corrected!
- While the associations between vitamin B12 status and liver function tests are adequately described, the discussion does not convey the clinical utility of the authors' findings. Why clinicians working in the field of GSDs should pay attention to elevated vitamin B12 levels? Are they associated with the degree of liver impairment? How these markers can potentially improve the assessment of these patients?
Response: Thank you for this suggestion. Although the mechanism underlying elevated plasma vitamin B12 and its possible implications in the long-term course of the disease require further investigation, we have addressed the clinical significance of our findings by adding a new paragraph to the discussion, as follows:
4.5. Clinical implications of elevated vitamin B12 in patients with hepatic GSD.
Elevated vitamin B12 concentrations are a common finding in hepatic GSD patients that may represent a previously unrecognized hallmark of this disease. Similar to elevated biotinidase activity found in most patients with hepatic GSD [65] plasma vitamin B12 may harbor diagnostic value. A combination of B12 measurements with other markers might improve the reliability of diagnosis. For instance, Table 6 shows that no single marker can separate the GSD-patients from the control subjects as good as the combinations of several markers (if working in a unidimensional space). Comparison of Fig. 4 vs Fig. 5 illustrates the same finding in a two-dimensional space (particularly visible for Fig. 4B vs Fig. 5C). The underlying mechanisms as well as the possible impact of metabolic control on plasma vitamin B12 concentration warrants further investigation. Long-term monitoring of vitamin B12 concentrations in the clinical course of patients may help to elucidate influencing factors, such as diet, metabolic control or medication.

Reviewer 2 Report
The authors in the manuscript entitled: "Vitamin B12 metabolism in hepatic glycogen storage diseases" for Journal of Clinical Medicine; present novel results that represent an advance in the knowledge of hepatic glycogen storage diseases. The article is written appropriately, the scientific design is good, although it would help readers' understanding and improve the value of the study. Therefore, it is suggested that the authors take into account, and change or modify the following aspects of the manuscript.
- Change the title to better suit the study. The study shows that patients with hepatic-GDS show an increase in the concentration of vitamin B12 in blood plasma. For this reason, would be more appropriate in view of the results, a title such as: Patients with hepatic glycogen disorders present high concentration of vitamin B12 than healthy subjects; or Vitamin B12 as a possible hallmarck in patients with hepatic-GSD.
- Change GDS for hepatic-GDS; since there are many more GDS that are not limited to the liver but to the muscle. Reading in the document only GDS induces to think at a systemic level
- The introduction is very concise. It would help to improve the manuscript markedly if the metabolism of vitamin B12 and its relevance at the liver level were described so that the reader is placed in the context in which the study focuses. Furthermore, it would also help if they mention the implications of vitamin B12 deficiency and accumulation in human physiology.
- In the material and methods section, the authors should describe the following sections in detail.
- settion 2.2 From which part is the blood drawn? How it is performed? What types of tube are used? Once the blood is in the tube with EDTA, explain the methodology in detail, centrifugation conditions, temperature, centrifuge model used, etc.
- Detailed description of the methodology in point 2.3 Explain in detail what the measurements of triacylglycerides, transaminase activity and how they are performed, indicating in each case the instruments used
- Brief description of point 2.4 Not only reference but briefly explain what it consists of and how it is done.
- In section 2.5 indicate the equations used to perform the calculations of the combined Vitamin B12 index.
- In the results section It is convenient and would substantially improve the manuscript if a box plot combined with a dot plot figure of the biochemical parameters determined were given healthy plasma versus hepatic-GDS. In such a way that a better disposition of the data and of the existing biological variability can be clearly observed. Like they have done in figure 2.
- Attach a graph correlating Vitamin B12 intake / plasma Vitamin B12 concentration that will provide a better understanding of the main finding of the manuscript (Plasma vitamin B12 concentration is significantly higher in 44 hepatic hepatic-GSD patients compared to healthy controls).
- Indicate the p-values of figure 2.
- Specify both in the figures and in the figure legend what type of measurements are the error bars standart deviation or standart error mean
- Figures 2 describe what is represented in the figure. The authors describe the results obtained; it is redundant because such information is reported in the results section
- To improve the study, it would be convenient to do studies of vitamin B12 transporters, and thus be able to make better speculations with the reason why patients with hepatic-GDS have higher levels of Vitamin B12 in plasma.
Author Response
The authors in the manuscript entitled: "Vitamin B12 metabolism in hepatic glycogen storage diseases" for Journal of Clinical Medicine; present novel results that represent an advance in the knowledge of hepatic glycogen storage diseases. The article is written appropriately, the scientific design is good, although it would help readers' understanding and improve the value of the study. Therefore, it is suggested that the authors take into account, and change or modify the following aspects of the manuscript.
Change the title to better suit the study. The study shows that patients with hepatic-GDS show an increase in the concentration of vitamin B12 in blood plasma. For this reason, would be more appropriate in view of the results, a title such as: Patients with hepatic glycogen disorders present high concentration of vitamin B12 than healthy subjects; or Vitamin B12 as a possible hallmarck in patients with hepatic-GSD.
Response: Thank you for this suggestion. We have now modified the title to more clearly illustrate the findings: ‘Elevated plasma vitamin B12 in patients with hepatic glycogen storage diseases’
- Change GDS for hepatic-GDS; since there are many more GDS that are not limited to the liver but to the muscle. Reading in the document only GDS induces to think at a systemic level
Response: We have indicated hepatic GSD clearly in the title of the manuscript and also in the description of the cohort under study. It is unlikely that readers will be confused in this respect. Given the high frequency of the use of the word ‘GSD’ in the manuscript, we believe adding ‘hepatic’ in every appearance of ‘GSD’ will be disruptive to the reading flow. The main text remains unchanged.
- The introduction is very concise. It would help to improve the manuscript markedly if the metabolism of vitamin B12 and its relevance at the liver level were described so that the reader is placed in the context in which the study focuses. Furthermore, it would also help if they mention the implications of vitamin B12 deficiency and accumulation in human physiology.
Response: Thank you. We have included a new paragraph describing these aspects, as follows:
Dietary vitamin B12 is carried into and through the digestive system via affinity-mediated binding to protein transporters, namely haptocorrin (HC), intrinsic factor (IF) and transcobalamin (TC)[4; 5]. Free, dietary vitamin B12 binds first to human HC present in the upper gastrointestinal tract (saliva and gastric juice). The vitamin B12-HC complex undergoes proteolysis in the lower portions of the intestine wherein the micronutrient then binds to IF, which is secreted by gastric parietal cells, but binds to B12 only upon normalization of pH. The complex of vitamin B12-IF is absorbed by ileal enterocytes, and within the enterocyte undergoes proteolysis and binding to the third vitamin B12 transporter, TC. Holo-TC enters portal circulation and is distributed to all cells in the body. Transcellular transport of vitamin B12 has been described for intestinal epithelial cells[6; 7] and vascular endothelial cells[8]. The organ with the greatest content of vitamin B12 is the liver followed by kidney and spleen[4]. As a result, certain liver diseases affect vitamin B12 status by influencing turnover and release of the micronutrient and its protein binders from hepatocytes into circulation.
In the material and methods section, the authors should describe the following sections in detail.
- settion 2.2 From which part is the blood drawn? How it is performed? What types of tube are used? Once the blood is in the tube with EDTA, explain the methodology in detail, centrifugation conditions, temperature, centrifuge model used, etc.
Response: Thank you for pointing to these missing methodological details. We have now modified the section as follows:
2.2. Handling of blood samples.
All venous blood samples were drawn from the arm and collected in EDTA-tubes (Sarstedt Monovette EDTA/KE 9 mL), centrifuged immediately at 4900 rpm (4168 x g), for 8 minutes at 4°C and stored at -80°C until analysis. Due to the risk of hypoglycaemia in patients with GSD, fasting time before blood draws was on average 3 hours, both for GSD patients and healthy controls. To permit for more reliable comparisons, the healthy control group also included arbitrary selected individuals whose blood was collected under non-fasting conditions.
- Detailed description of the methodology in point 2.3 Explain in detail what the measurements of triacylglycerides, transaminase activity and how they are performed, indicating in each case the instruments used
Response: Thank you once again for pointing to these missing methodological details. We have now modified the section as follows:
2.3. Determination of total vitamin B12, triglycerides and transaminases in plasma.
Total vitamin B12 concentrations were measured using an electrochemiluminescence immunoassay (Roche), triglycerides and transaminase activities were measured by routine techniques in the central diagnostic laboratory of the University Hospital Freiburg. Triglycerides and transaminases were analysed on a Cobas 8000 c502/C702 autoanalyser from ROCHE. Plasma vitamin B12 was analysed on a Cobas 8000 e802 autoanalyser from ROCHE.
2.3.1 GOT activity was measured at 37 oC according to the recommendations of the International Federation of Clinical Chemistry (IFCC). The GOT in the sample catalyzes the transfer of an amino group between L-aspartate and 2-oxoglutarate, producing oxaloacetate and L-glutamate. Oxaloacetate then reacts with NADH to form NAD + in the presence of malate dehydrogenase (MDH). Pyridoxal phosphate serves as a coenzyme in the amino transfer reaction, ensuring full enzyme activation. The rate of oxidation of NADH determined by decrease in absorbance at 340 nm is directly proportional to GOT activity. The linearity range is 5 – 700 U/L.
2.3.2 GPT activity was determined at 37 oC according to the guidelines of the IFCC, in the presence of pyridoxal phosphate. GPT catalyzes the transfer of the 2-amino group from alanine to 2-oxoglutarate to form glutamate and pyruvate. Formation of product pyruvate is followed by the coupled reaction of lactate dehydrogenase whereby NADH is oxidized to form NAD+. The consumption of NADH is monitored by measuring the absorbance at 340 nm, which is directly proportional to the rate of pyruvate formation by GPT activity. The linearity range is 5 – 700 U/L.
2.3.3. Serum triglycerides were determined by hydrolysis to glycerol and free fatty acids in a lipoprotein lipase-catalyzed reaction with subsequent oxidation to dihydroacetone phosphate and hydrogen peroxide. The formed hydrogen peroxide is quantified by the formation of a red dye by its reaction with 4-aminophenazone and 4-chlorophenol in the presence of peroxidase. This Trinder endpoint reaction has a linearity range of 8.85 – 885 mg/dl.
2.3.4 Total vitamin B12 concentrations were measured using a competitive electrochemiluminescence immunoassay with a calibration curve from 100 – 2000 pg/mL. The reference range of plasma B12 is 198-771 pg/ml. Vitamin B12 concentrations above 771 pg/mL were categorized as elevated vitamin B12.
- Brief description of point 2.4 Not only reference but briefly explain what it consists of and how it is done.
Response: Thank you. We have expanded the description of these biomarker determinations as follows:
Total plasma homocysteine concentrations were measured by tandem mass spectrometry as described earlier [15]. Briefly, 20 µL of plasma were mixed with 20 µL of DTT 0.5 M to reduce all free and protein-bound disulfides, vortexed and allowed to react at room temperature for 15 minutes. Twenty µL of internal standard D4-homocysteine (50 µM) were added and metabolites were extracted by addition of 100 µL of 0.1% formic acid in MeOH. The sample was centrifuged at 9447 × g for 10 min at room temperature and the resulting supernatants transferred into HPLC vials for LC-MS/MS analysis as described [15]. Methylmalonic acid levels in plasma were determined using liquid chromatography- tandem mass spectrometry as described elsewhere [16]. Briefly, 50 µL of plasma were mixed with 50 µL of internal standard D3-methylmalonic acid (0.8 µM) and sample cleanup was performed by ultrafiltration in a microcentrifuge tube. The filtrate was acidified with 10 µL of 4% formic acid and the sample transferred into an HPLC vial for subsequent LC-MS/MS determination as described [16]. Both for Hcy and MMA, assay performance quality was examined by incorporating a commercially available standardized marker for plasma analysis (Control special assays in serum, MCA, Product Nrs. SAS-02.1 and SAS-02.2).
- In section 2.5 indicate the equations used to perform the calculations of the combined Vitamin B12 index.
Response: Thank you. The equation was now added to section 2.5. along with an accompanying explanation.
The cB12 index was calculated using the following expression:
Here the first element of equation represents the combination of four markers of B12-status in the test sample, the second one reflects the value expected in a reference group (with correction for age), and the last element describes correction of folate-caused shift in Hcy. In brief, the index cB12 describes deviation of a test sample from a “normal” reference cohort, where cB12 around zero (or ≥ 0) indicates an adequate status, whereas negative values (e.g. -1, -2, -3) describe the grades of insufficiency (e.g. low B12, possible deficiency, probable deficiency).
- In the results section It is convenient and would substantially improve the manuscript if a box plot combined with a dot plot figure of the biochemical parameters determined were given healthy plasma versus hepatic-GDS. In such a way that a better disposition of the data and of the existing biological variability can be clearly observed. Like they have done in figure 2.
Response: The requested graphs have been added to supplemental information as Figure S1 to avoid redundancy of data presentation within Table 1.
- Attach a graph correlating Vitamin B12 intake / plasma Vitamin B12 concentration that will provide a better understanding of the main finding of the manuscript (Plasma vitamin B12 concentration is significantly higher in 44 hepatic hepatic-GSD patients compared to healthy controls).
Response: This new graph has been added to supplemental information as Figure S5.
- Indicate the p-values of figure 2.
Response: An asterisk was added to the figure and the p value is now provided in the Figure legend.
- Specify both in the figures and in the figure legend what type of measurements are the error bars standart deviation or standart error mean
Response: This information has been added to the Figure legend, as follows:
The middle line represents the 50th quartile (median). The lower and upper hinges represent the 25th and 75th quartiles, respectively. The distance between the 25th and the 75th quartiles depicts the inter-quartile range (IQR). The upper whisker represents the largest value that extends no further than 1.5 times the IQR. The lower whisker represents the smallest value that extends up to 1.5 times the IQR.
- Figures 2 describe what is represented in the figure. The authors describe the results obtained; it is redundant because such information is reported in the results section
Response: Thank you! With the clarification of the figure legend requested above, we now more clearly explain what is shown in the figure.
- To improve the study, it would be convenient to do studies of vitamin B12 transporters, and thus be able to make better speculations with the reason why patients with hepatic-GDS have higher levels of Vitamin B12 in plasma.
Response: We fully agree with the reviewer. This is already embedded in the discussion:
Plasma transporters of vitamin B12 (holo-TC and haptocorrin) and unsaturated cobalamin binding capacity (UCBC) [39] were not determined in this study. Therefore, it is currently unknown whether GSD patients with highly elevated vitamin B12 have a saturated cobalamin binding capacity of its authentic transport proteins. If so, excess vitamin B12 might bind to non-dedicated proteins such as albumin [40; 41; 42; 43] and the recently described immunocomplexes [44; 45; 46]. The biological activity of these protein complexes of vitamin B12 is unknown and currently under investigation.

Reviewer 3 Report
Summary: The authors have used a cohort of 44 patients with various glycogen storage disease (GSDs) and 42 healthy age- and gender-matched controls to see if there were differences in the vitamin B12 status. The rationale for this is linked to the previous findings that various pathological conditions, including liver diseases, lead to higher levels of vitamin B12 in the blood.
This study looks at vitamin B12 levels as well as two other metabolites that are associated with vitamin B12 metabolism (homocysteine and methylmalonic acid) to see how these GSDs change vitamin B12 status, as hypothesized. An effort was also made to see if vitamin B12 levels correlated with liver function, with triglyceride levels and the activities of two liver transaminases being measured.
Main concern/room for improvement:
- It is unclear how you were able to go from the questionnaire to actuallygiving a numerical estimate of vitamin B12 intake? Is there an equation that you can share or a citation to a paper that shows how this could be replicated? At the moment I’m sceptical of how quantitative your estimate can be just based on the questionnaire.
Also is it possible to add the units for vitamin B12 intake in Table 1.
- It is not very clear the significance of how the joint analysis of the different markers of liver function provided better separation of healthy and GSD patients. I think this needs to be explained more thoroughly, including the rationale for doing this combined analysis. At the moment the explanation that this just shows that “examining the metabolic landscape of patients” is “superior compared to the analysis of the individual tests” seems a little too vague. Not a huge concern but I feel a bit more thought into how all of these metabolites/enzymes work together would be helpful. Just a bit more thought into how this data may be useful.
- Some descriptions of the methods are very vague and would be impossible to try to replicate. For example, the measurement of triglycerides and transaminase activities is described simply as “by routine techniques in the central diagnostic laboratory of the University Hospital Freiburg”. Is it possible to add a citation to a paper that at least describes these methods?
Minor points:
- The interchanging of Cbl, “Vitamin B12” and “Vitamin B12” (with subscript) appears to be a little bit random, is it possible after acknowledging up front that Vitamin B12 is also called Cbl and then just using one term from then on? I’d suggest “Vitamin B12” as it’s in the title.
- The first sentence in section 3.4 starting with “The assessment of vitamin B12 status…” needs rewording, it was a little hard to follow.
- Seems a bit excessive having Table 5 and Figure 3 (essentially the showing the same thing twice).
- On page 10 the sentence “Such presentation suppresses inherent dispersion present in all individual markers” is unclear to me. Would you mind trying to rewrite I’m not quite sure what you mean.
- To be honest that whole paragraph on page 10 is a little hard to follow. For example the sentence “All the charts showed a significant correlation of the combined GSD-markers and B12, though not reaching the best correlation seen between B12 and GOT (Figure 4. Panel B. I’m not sure from the figure why this particular one was singled out as not a good correlation. Potentially it’s due to me not being too familiar with these Theil-Sin fits but if you could try making it a bit more clear it could help other readers like me.
- Are you sure you can definitively say “Elevated vitamin B12 cannot be unequivocally explained by liver dysfunction”? given that you only looked a small number of markers for liver dysfunction? I’d consider weakening that sentence.
- In the discussion you say “Although we did not find a correlation between vitamin B12 intake and plasma vitamin B12 concentrations in our study cohort…” I think you should say “estimated vitamin B12 intake) and also acknowledge that one problem could be that this was based of a fairly simple questionnaire that at best could only lead to an educated guess. For example, saying you eat beef 3 times a week could mean very different quantities to different people.
Author Response
Summary: The authors have used a cohort of 44 patients with various glycogen storage disease (GSDs) and 42 healthy age- and gender-matched controls to see if there were differences in the vitamin B12 status. The rationale for this is linked to the previous findings that various pathological conditions, including liver diseases, lead to higher levels of vitamin B12 in the blood.
This study looks at vitamin B12 levels as well as two other metabolites that are associated with vitamin B12 metabolism (homocysteine and methylmalonic acid) to see how these GSDs change vitamin B12 status, as hypothesized. An effort was also made to see if vitamin B12 levels correlated with liver function, with triglyceride levels and the activities of two liver transaminases being measured.
Main concern/room for improvement:
- It is unclear how you were able to go from the questionnaire to actually giving a numerical estimate of vitamin B12 intake? Is there an equation that you can share or a citation to a paper that shows how this could be replicated? At the moment I’m sceptical of how quantitative your estimate can be just based on the questionnaire.
Response: Thank you for pointing to the lack of this important detail in the estimation of vitamin B12 intake. We have clarified this in section 2.6 and provided the citation used to calculate B12 content in foods, as follows:
2.6. Estimation of vitamin B12 intake.
Oral vitamin B12 intake and possible additional supplementation were assessed using a questionnaire that addressed the patients’ nutritional habits. The questionnaire specifically focused on the frequency of consumption of vitamin B12-containing foods such as different types of meat, fish, eggs and dairy products, as well as on the supplementation with vitamin preparations containing vitamin B12. Vitamin B12 intake was estimated by multiplying the frequency of consumption by the average vitamin B12 content of these foods as given in the nutritional table published by the German Society of Nutrition (DGE)[19]. The questionnaire, evaluating Cbl intake, is provided in Supplemental Information (Figure S1).
Also is it possible to add the units for vitamin B12 intake in Table 1.
Response: The units of vitamin B12 intake have now been added in Table 1 (µg/day). Likewise, the missing units for vitamin B12 concentration in plasma were also added (pg/mL).
- It is not very clear the significance of how the joint analysis of the different markers of liver function provided better separation of healthy and GSD patients. I think this needs to be explained more thoroughly, including the rationale for doing this combined analysis. At the moment the explanation that this just shows that “examining the metabolic landscape of patients” is “superior compared to the analysis of the individual tests” seems a little too vague. Not a huge concern but I feel a bit more thought into how all of these metabolites/enzymes work together would be helpful. Just a bit more thought into how this data may be useful. B12
Response: We fully understand this point and appreciate the comment.
The rationale for the combined analysis of several GSD-markers (plus B12) was to explore diagnostic separation of the GSD cohort from the healthy subjects within a ‘mathematical continuum’. From the medical point of view, GSD patients (having elevated plasma B12 compared to healthy controls) are not just a prolongation of the normal range of plasma concentrations for B12 and GSD-markers (TG, GOT, GPT). In fact, GSD patients carry a discrete gene mutation that alters their metabolism in ways fully distinguishable from that of healthy subjects. Despite such predetermined difference, each single marker contains an inherent variability (“noise”). Simultaneous analysis of two or several combined markers in a two-dimensional space (like in Fig. 4 and Fig. 5) or a unidimensional space (like in Table 6) is expected to improve separation of the “affected” (positive) and “unaffected” (negative) cohorts by diminishing their overlap and minimizing the contribution of ‘noise’, as discussed in details elsewhere (S.N. Fedosov, Metabolic signs of vitamin B(12) deficiency in humans: computational model and its implications for diagnostics. Metabolism 59 (2010) 1124-38) and ref. [18]. Based on the aforementioned assumptions, we intended to critically examine the behavior of plasma B12 as a biomarker with respect to GSD-relevant metabolites (TG, GOT and GPT). This examination was driven by the long-standing curiosity of whether abnormally elevated plasma B12 could generically be attributed to liver dysfunction or if instead, liver biomarkers (and their combinations) could be exploited along with plasma B12 as a discrimination tool for ‘health versus GSD’. Not surprisingly, the analysis of merged datasets (healthy controls plus GSD patients) returned results different from those for each cohort taken separately. From this finding, one can predict the existence of a distinct vector that describes a ‘jump’ from one metabolic type to another (health to GSD, for example). Despite the artificial nature of this mathematical continuum, we succeeded in illustrating the power of simultaneous analysis of two or more markers, when separating healthy controls from GSD patients in the 2D-space (Figures 4 and 5)a and during the unidimensional analysis (Table 6). In both approaches, mutual overlap of two groups was lower, than when using only one marker.
To more clearly depict the motivation and results of this analysis, Figures 4 and 5 were updated, and section 3.7 was modified as follows:
3.7. Relationships between biomarkers upon merging healthy control and GSD groups as a continuum.
To further study the dependencies of the variables chosen in this study, we merged data from healthy controls and GSD patients. The motivation for this analysis was two-fold. First, we searched for a better diagnostic separation of the two groups under study (e.g. via a two-dimensional presentation of B12 vs an established GSD-marker). Secondly, we attempted to define the vector of metabolic transformation “healthy → GSD” within boundaries of two GSD-markers Y and X (including B12). In all cases, we aimed to test possible usability of B12 in GSD-diagnostics. Correlations between Y and X were examined by the Theil-Sen method, which estimates median slope and has a low sensitivity to outliers. The X,Y- and Y,X-fitting lines were calculated and used as a frame to draw the overall mean vector “healthy → GSD” together with a predicted 2D cut-off line, which produced the best separation between controls (red points) and GSD-patients (green points). The correlation between B12 and TG (Figure 4, panel A) was not statistically significant, yet the probability of equal medians for the cohorts of controls and patients was very low: p = 0.0002 (along X-axis) and p = 10−8 (along Y-axis), pointing to an association of the two variables. The approximate cut-off lines (yellow arrows in Figure 4) were drawn and gave reasonably low scores of the “misplaced points” (reds among greens / greens among reds): (7 / 4), (11 / 11), (10 / 11), (6 / 6) for panels A, B, C, D, respectively. Strictly vertical or horizontal cut-offs (based on a single variable, either X or Y) revealed higher overlaps. For example, panel D gave the counts (11 / 12) for the X-based cut off and (7 / 8) for the Y-based cut-off, both inferior to (6 / 6) obtained by 2D method with an angled separator (yellow double arrow). We next examined associations between vitamin B12 and distinct mathematical combinations of biomarkers relevant to GSD. Such combined analysis of biomarkers reduces the contribution of their individual variabilities thus permitting a more reliable assessment of their power to discriminate between healthy and diseased subjects, as discussed in another context in refs. [22; 23; 43]). Figure 5 explores logarithmic dependencies of B12 plotted versus several combinations of GSD-markers. A statistically significant correlation of the combined GSD-markers and B12 was identified in all cases (see p values in Figure 5), albeit not as strong as the correlation seen between B12 versus GOT (biomarkers not combined, Figure 4, panel B). Yet the panels in Figure 5 indicated a better separation of points for the cohorts of healthy controls and GSD patients. Thus, the counts of “misplaced” points (reds among greens / greens among reds) were assessed as (8 / 8), (6 / 5), (6 / 5) in panels A, B, C, respectively. These values are generally lower than those in Figure 4, where only two variables (one for each axis) were used. Probabilities of overlapping distributions were further assessed in a unidimensional space (with a simple or combined variable X), and the results are presented in Table 6. Optimal dissection of the two cohorts was obtained for ½·log10(GPT·TG), followed by ¼·log10(B12·GOT·GPT·TG) and â…“·log10(GOT·GPT·TG), according to the lowest probabilities of overlap. None of the individual markers came close to the effectiveness of the combined markers in discriminating the profiles of healthy controls and GSD patients.
Some descriptions of the methods are very vague and would be impossible to try to replicate. For example, the measurement of triglycerides and transaminase activities is described simply as “by routine techniques in the central diagnostic laboratory of the University Hospital Freiburg”. Is it possible to add a citation to a paper that at least describes these methods?
Response: Thank you for pointing this out. This missing information was also requested by Reviewer 2. We have now incorporated all necessary details in the manuscript, as follows:
2.3. Determination of total vitamin B12, triglycerides and transaminases in plasma.
Total vitamin B12 concentrations were measured using an electrochemiluminescence immunoassay (Roche), triglycerides and transaminase activities were measured by routine techniques in the central diagnostic laboratory of the University Hospital Freiburg. Triglycerides and transaminases were analysed on a Cobas 8000 c502/C702 autoanalyser from ROCHE. Plasma vitamin B12 was analysed on a Cobas 8000 e802 autoanalyser from ROCHE.
2.3.1 GOT activity was measured at 37 oC according to the recommendations of the International Federation of Clinical Chemistry (IFCC). The GOT in the sample catalyzes the transfer of an amino group between L-aspartate and 2-oxoglutarate, producing oxaloacetate and L-glutamate. Oxaloacetate then reacts with NADH to form NAD + in the presence of malate dehydrogenase (MDH). Pyridoxal phosphate serves as a coenzyme in the amino transfer reaction, ensuring full enzyme activation. The rate of oxidation of NADH determined by decrease in absorbance at 340 nm is directly proportional to GOT activity. The linearity range is 5 – 700 U/L.
2.3.2 GPT activity was determined at 37 oC according to the guidelines of the IFCC, in the presence of pyridoxal phosphate. GPT catalyzes the transfer of the 2-amino group from alanine to 2-oxoglutarate to form glutamate and pyruvate. Formation of product pyruvate is followed by the coupled reaction of lactate dehydrogenase whereby NADH is oxidized to form NAD+. The consumption of NADH is monitored by measuring the absorbance at 340 nm, which is directly proportional to the rate of pyruvate formation by GPT activity. The linearity range is 5 – 700 U/L.
2.3.3. Serum triglycerides were determined by hydrolysis to glycerol and free fatty acids in a lipoprotein lipase-catalyzed reaction with subsequent oxidation to dihydroacetone phosphate and hydrogen peroxide. The formed hydrogen peroxide is quantified by the formation of a red dye by its reaction with 4-aminophenazone and 4-chlorophenol in the presence of peroxidase. This Trinder endpoint reaction has a linearity range of 8.85 – 885 mg/dl.
2.3.4 Total vitamin B12 concentrations were measured using a competitive electrochemiluminescence immunoassay with a calibration curve from 100 – 2000 pg/mL. The reference range of plasma B12 is 198-771 pg/ml. Vitamin B12 concentrations above 771 pg/mL were categorized as elevated vitamin B12.
2.4. Determination of tHcy and MMA.
Total plasma homocysteine concentrations were measured by tandem mass spectrometry as described earlier [15]. Briefly, 20 µL of plasma were mixed with 20 µL of DTT 0.5 M to reduce all free and protein-bound disulfides, vortexed and allowed to react at room temperature for 15 minutes. Twenty µL of internal standard D4-homocysteine (50 µM) were added and metabolites were extracted by addition of 100 µL of 0.1% formic acid in MeOH. The sample was centrifuged at 9447 × g for 10 min at room temperature and the resulting supernatants transferred into HPLC vials for LC-MS/MS analysis as described [15]. Methylmalonic acid levels in plasma were determined using liquid chromatography- tandem mass spectrometry as described elsewhere [16]. Briefly, 50 µL of plasma were mixed with 50 µL of internal standard D3-methylmalonic acid (0.8 µM) and sample cleanup was performed by ultrafiltration in a microcentrifuge tube. The filtrate was acidified with 10 µL of 4% formic acid and the sample transferred into an HPLC vial for subsequent LC-MS/MS determination as described [16]. Both for Hcy and MMA, assay performance quality was examined by incorporating a commercially available standardized marker for plasma analysis (Control special assays in serum, MCA, Product Nrs. SAS-02.1 and SAS-02.2).
Minor points:
- The interchanging of Cbl, “Vitamin B12” and “Vitamin B12” (with subscript) appears to be a little bit random, is it possible after acknowledging up front that Vitamin B12 is also called Cbl and then just using one term from then on? I’d suggest “Vitamin B12” as it’s in the title.
Response: Thank you. We have now unified the use of abbreviations, retaining ‘vitamin B12’ throughout the manuscript.
- The first sentence in section 3.4 starting with “The assessment of vitamin B12 status…” needs rewording, it was a little hard to follow.
Response: Thank you. We have reworded this sentence so it now reads as follows:
The reliable assessment of vitamin B12 status requires the measurement of two to four biomarkers.
- Seems a bit excessive having Table 5 and Figure 3 (essentially the showing the same thing twice).
Response: We agree with the reviewer. Table 5 was moved into Supplemental Information as Table S1.
- On page 10 the sentence “Such presentation suppresses inherent dispersion present in all individual markers” is unclear to me. Would you mind trying to rewrite I’m not quite sure what you mean.
Response: We agree this sentence needed rewording to improve clarity. All biological parameters present dispersion with respect to the calculated mean or median of the entire population. This dispersion is associated with both analytical imprecision and variations in metabolic fluxes. For example, the level of any metabolite in blood is determined by at least three independent fluxes (production / intake, consumption and excretion), while the disease usually affects only one of them. The examination of several biomarkers in combination with one another, e.g. via multiplication of their values or/and presentation on a 2D-surface is one available strategy to reduce the impact of such deviations. We tried to communicate this statement as follows:
We next examined associations between vitamin B12 and distinct mathematical combinations of biomarkers relevant to GSD. Such combined analysis of biomarkers reduces the contribution of their individual variabilities, thus permitting a more robust discrimination between healthy and diseased subjects (discussed in another context in refs. [22; 23]).
To be honest that whole paragraph on page 10 is a little hard to follow. For example the sentence “All the charts showed a significant correlation of the combined GSD-markers and B12, though not reaching the best correlation seen between B12 and GOT (Figure 4. Panel B. I’m not sure from the figure why this particular one was singled out as not a good correlation. Potentially it’s due to me not being too familiar with these Theil-Sin fits but if you could try making it a bit more clear it could help other readers like me.
Response: We appreciate the comment and the request for clarification, which will by all means help our readers. We failed to made it clear why we chose to look for correlations using mathematical expressions that combine data of biomarkers, instead of using them in the more traditional manner, i.e. one variable versus one other variable. We have now reworded that first portion of the paragraph and together with the edits above, we hope our data analysis becomes more intuitive to follow:
A statistically significant correlation of the combined GSD-markers and B12 was identified in all cases for ‘healthy ®GSD’ vectors (see p values in Figure 5), albeit not as strong as the correlation seen between B12 versus GOT (Figure 4, panel B, the lowest p, biomarkers not combined).
- Are you sure you can definitively say “Elevated vitamin B12 cannot be unequivocally explained by liver dysfunction”? given that you only looked a small number of markers for liver dysfunction? I’d consider weakening that sentence.
Response: We hypothesized that liver disease in GSD patients may be the underlying cause of elevated plasma vitamin B12. However, we found no associations between plasma vitamin B12 and liver enzymes GOT and GPT, two biomarkers of liver physiology. It should be noted that this observation is true for the separated cohort of healthy controls and GSD patients. The overall vector ‘healthy® GSD‘ indicated a strong correlation upon a jump from ‘healthy’ to ‘GSD’. We also agree that these are only two markers that do not holistically describe liver function. Therefore, we have reworded this sentence as follows:
Elevated plasma vitamin B12 does not associate with liver function biomarkers GOT and GPT in our cohort of patients with GSD, even though the overall vector ‘healthy® GSD’ showed a correlation upon a ‘jump’ from one metabolic state to another.
- In the discussion you say “Although we did not find a correlation between vitamin B12 intake and plasma vitamin B12 concentrations in our study cohort…” I think you should say “estimated vitamin B12 intake) and also acknowledge that one problem could be that this was based of a fairly simple questionnaire that at best could only lead to an educated guess. For example, saying you eat beef 3 times a week could mean very different quantities to different people.
Response: We fully agree that this is a weakness of our study. We rewrote that section as follows:
Although we did not find a correlation between the estimated vitamin B12 intake and plasma vitamin B12 concentrations in our study cohort, elevated vitamin B12 levels were more common in ketotic GSD patients compared to GSD I patients (50% versus 22%). This could be due to a distinct effect of GSD on plasma vitamin B12 concentration or due to statistical bias introduced by the small size of the cohort examined herein. Besides, we acknowledge that the estimated vitamin B12 intake determined via our nutrition questionnaire only allows for an educated guess of this variable as opposed to the accuracy that is achieved via standardized dietary intake protocols.

Round 2
Reviewer 2 Report
The authors in the manuscript jcm-857039 for Journal of Clinical Medicine. After a first round of review, they present results, as suggested. This has led to a noticeable improvement in the text. Enabling the reader to have a better description and understanding of the study presented. For this reason, I consider that the manuscript is suitable to publish in the Journal of Clinical Medicine.